# Rise of the war machines: Charting the evolution of military technologies from the Neolithic to the Industrial Revolution

**Peter Turchin**[1,2,3], **Daniel Hoyer**[4,5]\*, **Andrey Korotayev**[6,7], **Nikolay Kradin**[8], **Sergey Nefedov**[9], **Gary Feinman**[10], **Jill Levine**[4], **Jenny Reddish**[1], **Enrico Cioni**[3], **Chelsea Thorpe**[3], **James S. Bennett**[11], **Pieter Francois**[3], **Harvey Whitehouse**[3]

**1** Complexity Science Hub, Vienna, Austria, **2** University of Connecticut, Storrs, Connecticut, United States of America, **3** University of Oxford, Oxford, England, **4** Evolution Institute, Tampa, FL, United States of America, **5** George Brown College, Toronto, Canada, **6** HSE University, Moscow, Russia, **7** Institute of Oriental Studies, Russian Academy of Sciences, Moscow, Russia, **8** Institute of History, Archaeology and Ethnology, Far East Branch of the Russian Academy of Sciences, Vladivostok, Russia, **9** Institute of History and Archeology of the Ural Branch of the Russian Academy of Sciences, Ural Federal University, Yekaterinburg, Russia, **10** Field Museum of Natural History, Chicago, IL, United States of America, **11** University of Washington, Seattle, Washington, United States of America

\* dhoyer@evolution-institute.org

**Data Availability Statement:** All relevant data are within the manuscript and its Supporting Information files. Additionally, all data are available

## Abstract

What have been the causes and consequences of technological evolution in world history? In particular, what propels innovation and diffusion of military technologies, details of which are comparatively well preserved and which are often seen as drivers of broad socio-cultural processes? Here we analyze the evolution of key military technologies in a sample of pre-industrial societies world-wide covering almost 10,000 years of history using *Seshat: Global History Databank*. We empirically test previously speculative theories that proposed world population size, connectivity between geographical areas of innovation and adoption, and critical enabling technological advances, such as iron metallurgy and horse riding, as central drivers of military technological evolution. We find that all of these factors are strong predictors of change in military technology, whereas state-level factors such as polity population, territorial size, or governance sophistication play no major role. We discuss how our approach can be extended to explore technological change more generally, and how our results carry important ramifications for understanding major drivers of evolution of social complexity.

## Introduction

From simple sharpened stone projectiles in the Paleolithic to the weapons of mass destruction in the modern world, what have been the main factors driving the evolution of military technology? Many have argued that the evolution of military technologies is just one aspect of a much broader pattern of technological evolution driven by increasing size and interconnectedness among human societies [1–3]. Several cultural evolutionary theories, conversely, highlight

alongside a preprint publication of this article, accessible at: https://osf.io/mkhde/

**Funding:** This work was supported by: a John Templeton Foundation grant to the Evolution Institute, entitled "Axial-Age Religions and the Z-Curve of Human Egalitarianism" (HW, PF, PT); a Tricoastal Foundation grant to the Evolution Institute, entitled "The Deep Roots of the Modern World: The Cultural Evolution of Economic Growth and Political Stability" (PT); an Economic and Social Research Council Large Grant to the University of Oxford, entitled "Ritual, Community, and Conflict" (REF RES-060-25-0085) (HW); a grant from the European Union Horizon 2020 research and innovation programme (grant agreement No 644055 [ALIGNED, www.aligned-project.eu]) (HW, PF); a European Research Council Advanced Grant to the University of Oxford, entitled "Ritual Modes: Divergent modes of ritual, social cohesion, prosociality, and conflict" (HW, PF); a grant from the Institute of Economics and Peace to develop a Historical Peace Index (HW, PF, PT, DH); and the program "Complexity Science," which is supported by the Austrian Research Promotion Agency FFG under grant #873927 (PT).

**Competing interests:** The authors have declared that no competing interests exist.

military technologies as a special case, arguing that steep improvements in both offensive and defensive capabilities of technologies along with accompanying tactical and organizational innovations resulted in "Military Revolutions" (note the plural), which in turn had major ramifications on the rise and, of particular concern here, the spread of state formations globally [4–8] and the evolution of religion and other cultural phenomena [9,10]. But the evolutionary mechanisms underlying general technological innovation, adoption, and transmission (especially in pre-industrial societies) are not well understood. Moreover, available theories have drawn on evidence that is limited both in geographical scope and temporal depth and deployed in ways that are subject to selection bias. Here we explore a variety of factors that previous scholarship suggests may have played a role in the evolution of military technologies by systematically quantifying the effects of those factors for thousands of years of world history.

Earlier efforts to quantify levels of technological complexity in eastern and western ends of Eurasia [11,12] have been criticized for being unduly subjective [13], especially when it comes to measuring rates of innovation in military technology, and are obviously limited in spatial coverage. Here we propose an alternative methodology for quantifying technological evolution and expand the geographic scope from just these two broad regions to 35 "Natural Geographic Areas" across all ten major world regions, using *Seshat*: *Global History Databank*, a major resource for studying patterns of sociocultural evolution in world history (see *Materials and Methods* below).

This article has two related goals. The first is to establish broad spatio-temporal patterns in the evolution of military technologies in pre-industrial societies. By technological evolution we mean here the dynamics of uptake (and possible loss) of technologies used by societies at significant scale (rather than simply whether the technology was known at all), regardless of how that society came to acquire that technology (indigenous innovation or adoption from another culture). For those interested in the study of technological evolution in general, focusing specifically on military technologies in pre-industrial societies has many practical benefits. Warfare was one of the most intensive activities of human societies, leaving abundant traces in the archaeological and historical record.

The second goal is to explore why these important military technologies developed or were adopted in the places, at the times, and as part of the technological packages as we observe in the historical and archaeological record. There have been several theoretical conjectures (discussed below) about the main causal drivers of technological innovation that we test. As our approach will show, the pattern of military technological evolution shows great variation in time and space, with different regions assuming a leading role in innovation at different moments in time.

Delineating the possible causes and observed consequences of changes in levels of military technologies will have far-reaching implications for understanding the evolution of technology broadly. To encourage further progress towards that ultimate goal, we present here a detailed methodology for testing theories about technological change in human history. This paper serves as a crucial step along this path.

## Theoretical background

Here, we review several competing theoretical perspectives on the evolution of technologies offered in the past. Technological change is one of the fundamental drivers in social and cultural evolution and of long-term economic growth [14–17]. Many have pointed to technology's ramifying effects on warfare, state formation, and the development of information processing systems [1,18–22]. Military technologies and their widespread application in particular have been shown to foment rises in social complexity and to spur related ideological

developments [23–27]. But what processes are responsible for the evolution–the development, spread, and cumulative adoption–of military technology globally and across time?

Following this link between military technologies and socio-cultural development, we might expect to find a positive feedback between technological innovation and population growth at the global scale [2,28–32] see also [33–36]. Indeed, a well-known and much discussed theory proposed by economist Michael Kremer and expanded by others suggests exactly this causal link [2]. According to Kremer, "high population spurs technological change because it increases the number of potential inventors." Kremer notes that "this implication flows naturally from the nonrivalry of technology. . . The cost of inventing a new technology is independent of the number of people who use it. Thus, holding constant the share of resources devoted to research, an increase in population leads to an increase in technological change. Thus, in a larger population there will be proportionally more people lucky or smart enough to come up with new ideas" [2: 685]. This innovation, in turn, can spur further population growth, creating a positive feedback loop between technological and population growth; for instance, the proliferation of iron axes facilitated the clearing of agricultural land from forests [37], while the iron ploughshare improved the quality of plowing allowing for increased productivity [38] and, hence, larger populations to develop further innovations. Note that what is described by Kremer is virtually identical with what David Christian calls "collective learning" [39].

This process was expressed mathematically by Taagepera, Kremer, Podlazov and Tsirel in the following way:

$$\frac{dT}{dt} = kNT \tag{1}$$

This equation states that the technological growth rate at a given moment in time ($dT/dt$) is proportional to the global population, $N$ (the larger the population, the larger the number of potential inventors) and to the current technological level, $T$. The second factor is included in the model to reflect the assumption that the wider the existing technological base, the greater the number of inventions that can be made on its basis. This model explicitly refers to global population level, rather than regional or localized populations of specific societies. To account both for the effect of global population size and the existing stock of technology, the Taagepera-Kremer model assumes that the rate of technology growth is proportional to the product of these two quantities. Taagepera and Kremer did not test this hypothesis empirically in a direct way. Empirical tests of this hypothesis performed by other researchers, however, have found support [34,40]. Note that Kremer observed that these new technologies would, in turn, likely generate population growth, suggesting a positive feedback between technological innovation and population. Here, however, we are concerned only with the effect of population on the evolution of technology, rather than the reverse.

A limitation of such population-focused theories, however, is the assumption that world population can be treated as having been an integrated information-exchanging system for many centuries, if not millennia. To address this problem, world-systems analysts have advanced an additional cluster of hypotheses. Chase-Dunn and Hall, for example, distinguish four types of networks of world-system communications: bulk good networks, political-military networks, prestige good networks, and information networks (IN) [41]. Korotayev et al. [42–44] explicitly focus on INs as technological innovation diffusion networks, proposing that a systematic diffusion of technological innovations within a certain set of societies is a sufficient condition to consider them a "world-system". Thus, an as-yet unexplored synthesis of these ideas is that, while population may be one factor in the pace and location of technological evolution, membership in such an information diffusion network may play an additional role

in facilitating the exchange of ideas and propensity for wide-spread adoption of new technologies.

One important advantage of the population-driven model advocated by Kremer, Taagepera, and others is that it explicitly includes the effect of the existing stock of technologies on technological growth rate. The greater the existing stock, the greater number of new technologies the model expects to be developed in the next time period. Although this is only one, relatively straightforward, way to model the impact of the existing technology stock, there is substantial historical evidence to make it a strong contender to be tested empirically. For example, the improvement of metallurgy and metal processing led not only to the emergence of new tools such as iron ploughs, but also to the proliferation of various types of weapons—starting with knives, daggers, swords, battle axes, up to the appearance of rifles and artillery. Nevertheless, the model assumes that the means and knowledge to adapt and improve upon existing technologies are readily accessible as well as the organizational capacity to deploy these technologies at large scales, which are open questions requiring further scrutiny.

Further, once a military technology had proven advantageous in inter-state competition, there arose an existential pressure on nearby societies to adopt that technology as well, so as not to be left behind. This sort of mimetic diffusion has been observed with respect to key technologies such as horse-mounted warfare that spread initially among nomadic confederations and nearby agrarian societies located along the central Eurasian Steppe [45–48]. Indeed, the domestication of the horse and its use in the civil and military sphere–including both the material components of horse-mounted archery as well as the tactical and organizational means to wield these weapons–appear to be of particular importance in the evolution of technologies and social complexity during the pre-industrial era, improving transportation, agriculture, and military capacities alike [47]. Further, the creation of new and more lethal weapons in one society could force people in their "strike zone" [27] to invent more sophisticated defenses while also often adopting the offensive technology themselves, prompting further technological advances. Following the invention of increasingly powerful, armor-piercing projectiles from bows and crossbows, for instance, we tend to see the means of protection improved as well to include chain mail, scaled armor, and plate armor.

Similarly, some work suggests that location is a critical factor in this process, as societies on the periphery, or semi-periphery [41], of larger, more complex imperial states will tend to be hotbeds of innovation, as they have both the incentive to increase (typically military) capability to compete with regional powers as well as the requisite flexibility to explore more radical innovation by being removed from the institutionalized practices and path-dependencies experienced by the larger societies "locked in" to the tools and habits that won them their hegemony [41,49,50].

Overall, previous theoretical work suggests that the evolution of military technologies depends on the total number of potential innovators involved in this process, the connectedness of distinct centers of innovation as well as of spheres of inter-state competition, and on the already existing stock of technologies, especially such fundamental developments as metal processing and transportation. In *Materials and Methods* below we discuss how we operationalize an empirical test of these hypotheses.

## Materials and methods

### A general approach to quantifying the evolution of pre-industrial societies

This article follows the general philosophy and procedures that have been developed by the *Seshat*: *Global History Databank* project [51–54]. The Seshat Databank stores large volumes of historical and archaeological data on a growing number of variables for past polities going

back to the late Neolithic. Supplementary Information (*SI*) contains a detailed description of the core methods and workflows underpinning the Seshat project, including how we incorporate differing levels of uncertainty and disagreement and data quality procedures involving experts and research assistants. We make the data used for the analyses presented here available online through a DataBrowser site (seshatdatabank.info/databrowser) and we encourage scholars to make use of and to augment our dataset.

The principal unit for data collection and analysis is a polity, defined as any independent political unit ranging from autonomous villages (local communities) through simple and complex chiefdoms to states and empires, regardless of degree of centralization [51,52]. Our sample of historic polities was developed using a stratified sample of the globe using the concept of 'Natural Geographic Area' (NGA). An NGA is a fixed spatial location of roughly 100 x 100 km delimited by naturally occurring geographical features (for example, a river basin, a coastal plain, a mountain valley, or an island). All polities that occupied the NGA, or part thereof, at a century mark (e.g. 200 CE), are included in our sample. This strategy avoids oversampling (redundantly repeating information across time points) while still capturing meaningful changes in the variables of interest. Although this granularity is relatively coarse, it is suitable for uncovering macro-level patterns in societal dynamics and exploring pathways of cultural evolution [24,53]. The data used in the analyses presented here come from 373 historic polities covering 35 NGAs.

## Aggregation of military technology data into "Warfare Characteristics"

We quantified the sophistication of war-making capacity by encoding 46 binary variables indicating the presence or absence of different military technologies by a polity. These variables were aggregated into six general categories, termed Warfare Characteristics (WCs): Metals used in producing weapons and armor, the variety of Projectiles and hand-held Weapons, the sophistication of Armor, the use of transport Animals, and different kinds of Defensive Fortifications. Finally these WCs were aggregated into a composite, temporal MilTech variable for each NGA. See *SI* for details of the aggregation.

Of the six WCs, two (Metal and Animal) have a much broader area of application than specifically warfare. In some analyses below we investigate another measure (CoreMil) that focuses more narrowly on the sophistication of core military technologies by aggregating only the Projectiles, Weapon, Armor, and Defense WCs. As described below, we explore the impact of the spread of Iron and Cavalry in particular. Because Iron and Cavalry are correlated with the Metals and Animals WCs, analyzing CoreMil allows us to disentangle any potentially spurious effects of these WCs on overall military technology.

## Hypotheses to be tested: Defining predictor variables

Our review in *Theoretical Background* suggested that the evolution of military technologies may be a function of the total number of potential innovators, the connectedness of innovation/adoption centers, and/or the existing stocks of technology. We measure these various potential explanatory factors in the following ways:

Following Taagepera and Kremer, we proxy the number of potential innovators with the world population (WorldPop, defined as log(10) of the global population at time *t*). We take data on the dynamics of world population during the Holocene from [55].

Connectedness is a harder variable to quantify. Here we build on the concept of IN used by Chase-Dunn and Hall and other world-systems theorists [34,43] who define the extent of any particular IN as the zone within which spatially and culturally distinct regions exchange information, so that technological innovations made in one society diffuse relatively rapidly (on the

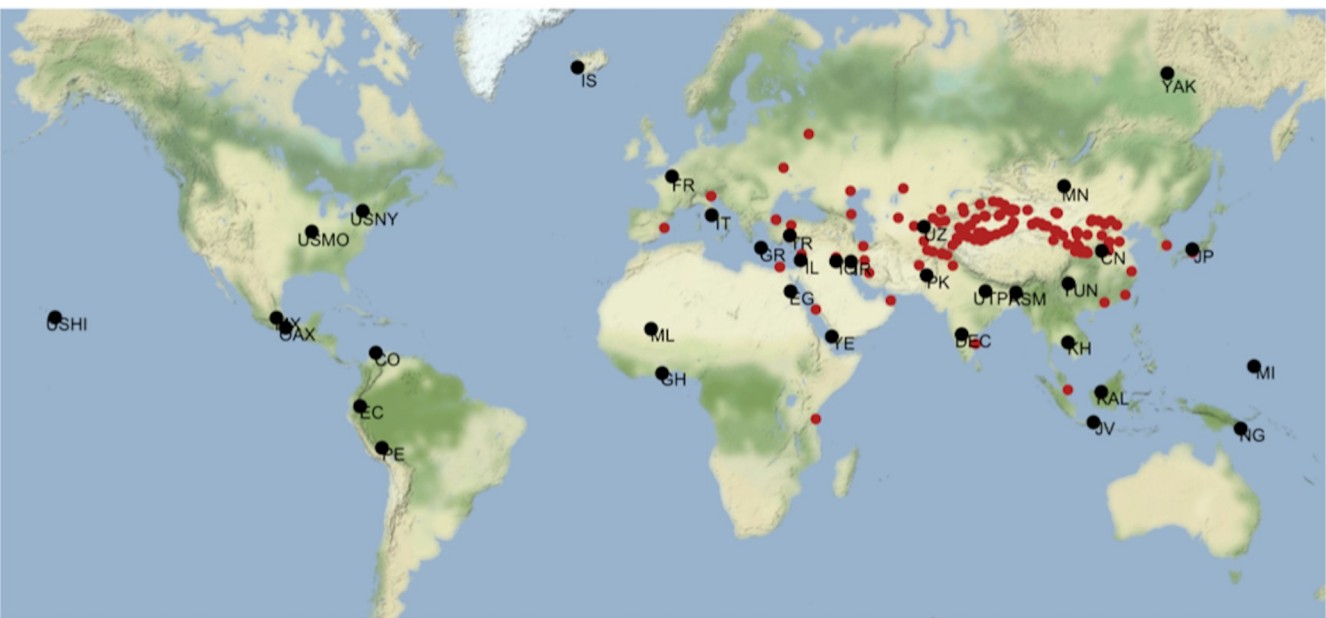

**Fig 1. Location of nodes on Silk Routes used in quantifying Centrality (red) along with NGA locations (black).**

time scale of centuries) to all other societies within the system than to societies that may be close (spatially and culturally) but fall outside of the IN. As an example, the contacts between Western and Eastern Eurasia (mediated via Central Asia) in the third and especially the second millennia BCE led to the spread of multiple technological innovations between the western and eastern parts of Eurasia: wheat, cattle, horses, bronze metallurgy, wheeled chariots, among others [44,56]. Here, we constructed a predictor variable proxying the Centrality of each region within the evolving (eventually global) IN by calculating the distance between each of our NGAs and the system of Silk Routes that connected East and West Eurasia for the majority of the period under study [57–60]. Our measure of Centrality is the inverse of the distance between an NGA and the nearest node on the Silk Route (see Fig 1; and *SI* for details).

In addition to Centrality within the IN, we capture two additional kinds of connectivity, namely the possible influence of spatial proximity (Space) as well as cultural affinity (Phylogeny). These terms not only allow us to control for possible autocorrelations and phylogenetic effects in our response variable (see *Dynamic Regression Analysis* below), but can also carry important information about processes influencing the evolution of military technologies. In particular, Space captures the process by which technological innovations may travel between geographically proximate societies–separately from the possible mediating influence of an expanding IN described above–measuring the likelihood that neighboring regions will share similar levels of military technology. Phylogeny focuses on the cultural similarity between polities, however spatially close, proxied by the relatedness of their dominant languages.

Another possible factor in the evolution of technology identified in the theoretical review is the effect of the current technology stock. We measure this in two ways. First, we model Mil-Tech as a temporal autoregressive process, in which past values of MilTech affect its future values (for the details of the statistical model, see the next section). Second, we focus on the potential effects of two specific fundamental technologies: horse-riding and iron smelting.

According to the Cavalry Revolution theory, the invention of effective horse-riding in the Pontic-Caspian steppes, combined with powerful recurved bows and iron-tipped arrows,

| NGA | Region | NGACode |
|---|---|---|
| Basin of Mexico | Mexico | MX |
| Big Island Hawaii | Hawaii | USHI |
| Cahokia | Illinois | USMO |
| Cambodian Basin | Cambodia | KH |
| Central Java | Indonesia | JV |
| Chuuk Islands | Micronesia | MI |
| Crete | Greece | GR |
| Cuzco | Peru | PE |
| Deccan | Deccan | DEC |
| Finger Lakes | New York | USNY |
| Galilee | Levant | IL |
| Garo Hills | Assam | ASM |
| Ghanaian Coast | Ghana | GH |
| Iceland | Iceland | IS |
| Kachi Plain | Pakistan | PK |
| Kansai | Japan | JP |
| Kapuasi Basin | Malaysia | KAL |
| Konya Plain | Turkey | TR |
| **NGA** | **Region** | **NGACode** |
| Latium | Italy | IT |
| Lena River Valley | East Siberia | YAK |
| Lowland Andes | Ecuador | EC |
| Middle Ganga | Uttar Pradesh | UTPR |
| Middle Yellow River Valley | Henan | CN |
| Niger Inland Delta | Mali | ML |
| North Colombia | Colombia | CO |
| Orkhon Valley | Mongolia | MN |
| Oro PNG | New Guinea | NG |
| Paris Basin | France | FR |
| Sogdiana | Uzbekistan | UZ |
| Southern China Hills | Yunnan | YUN |
| Southern Mesopotamia | Iraq | IQ |
| Susiana | Iran | IR |
| Upper Egypt | Egypt | EG |
| Valley of Oaxaca | Oaxaca | OAX |
| Yemeni Coastal Plain | Yemen | YE |

triggered a process of military evolution that spread from the steppes south to the belt of farming societies over several centuries throughout the first millennia BCE and CE [8,47,61]. Specifically, the threat of nomadic warriors armed with this advanced (for the period) military technology spurred the development of counter-measures designed to mitigate the cavalry advantage, while also producing an incentive to adopt horse-riding and effective accompanying combat tactics in areas further and further away from the location of their initial invention within the Steppe. The history of the military use of the horse went through several stages: the use of the chariot, the development of riding, the formation of light auxiliary cavalry, the development of nomadic riding, the appearance of the hard saddle, armored cataphracts, stirrups and, finally, heavy cavalry—a major branch of troops across Afro-Eurasian societies

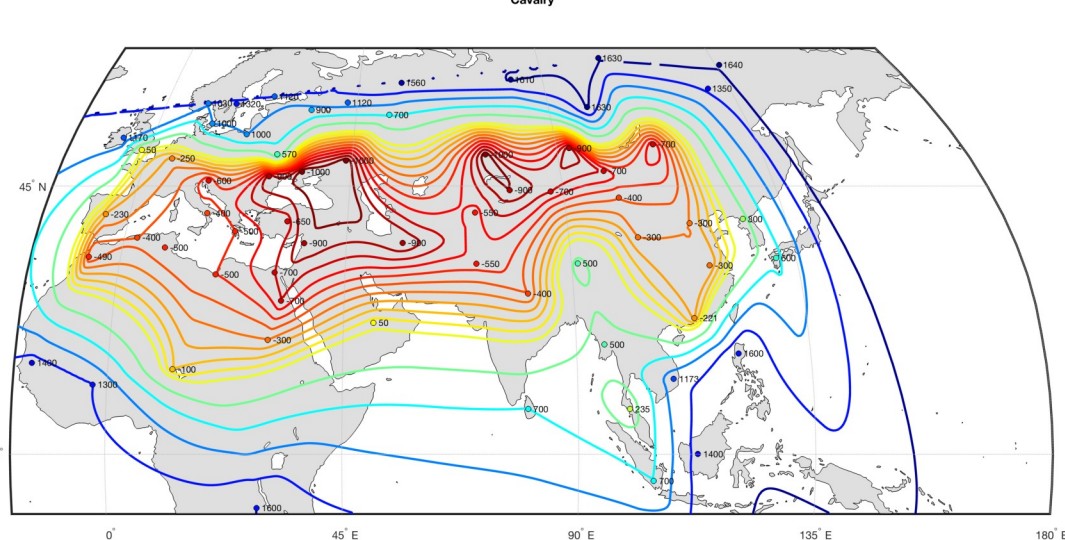

**Fig 2. Spread of horse-mounted Cavalry.** Data from [63].

between c. 550 and 1400 CE [62]. As a result, effective horse-riding had far-reaching consequences for the evolution of military technologies, and specifically armor, projectiles such as crossbows, and fortifications. We use the data from [63] to encode the Cavalry variable (see Fig 2).

The effect of Iron is similarly widespread. Multiple authors [64–66] have suggested that the availability of iron had a major impact on the evolution of technologies, as this strong and malleable material served as an input for a host of important technologies, military and otherwise, throughout the period under investigation here. We use data from [67] to encode the Iron variable (see Fig 3).

Note that these two variables, Cavalry and Iron, are highly correlated with each other (compare Figs 2 and 3) and it may be difficult to estimate their effects separately (the problem of

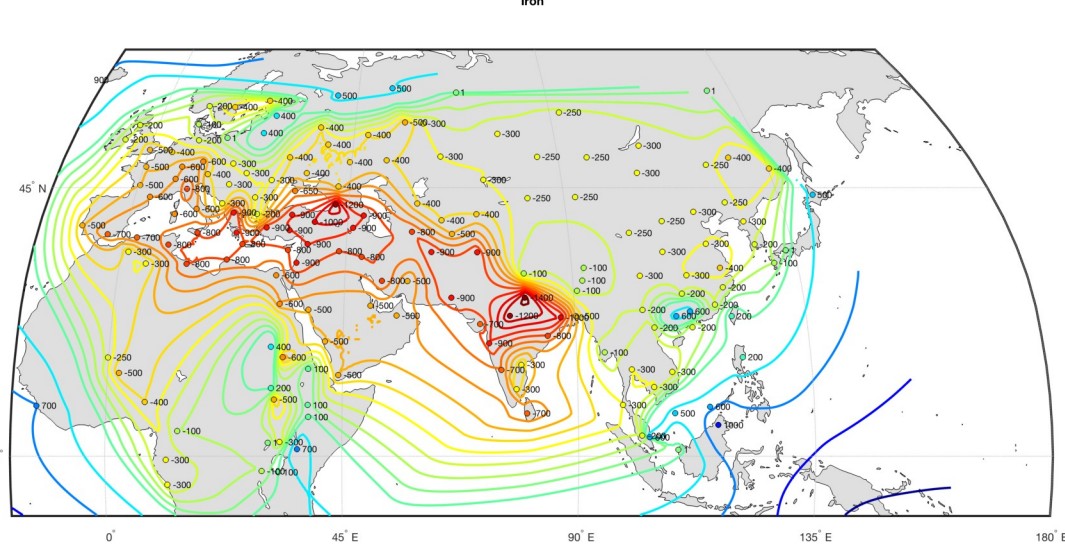

**Fig 3. Spread of iron metallurgy.** Data from [67].

collinearity). To address this potential issue we created a synthetic variable, IronCav, that combines the two effects (by adding Cavalry and Iron together). IronCav, thus, takes the maximum value for societies with both mounted warfare and iron weapons, intermediate value for societies having one characteristic and not the other, and minimum for societies with neither characteristic. We explored with dynamic regressions whether IronCav turns out to be a better predictor than either of its constituent variables, reported below.

In addition to the theoretically-motivated predictors–WorldPop, Centrality, Iron, and Cavalry, along with our autocorrelation terms Space and Phylogeny–we explore other potential polity- and NGA-specific predictors to proxy interesting subsidiary hypotheses, as explained below. These measures are taken from previously published analyses using Seshat data [68] and enable us to reduce the potential "hidden variable" (or omitted variable bias) problem, which arises when analysis implicates *X* as a causal factor for *Y*, while in reality the true cause could be *Z*, with which *X* is closely correlated [69,70]. The additional predictor variables include the following:

1. Social scale (Scale) represents the first principal component (PC) of the Seshat variables *polity population*, *polity territory*, *population of the largest settlement*, and *the number of hierarchical levels*. The hypothesis here is that larger and more complexly organized and productive societies (in both population, territory) will have more resources to both generate new inventions and to implement them, or adopt them from elsewhere, especially the costly ones like sophisticated siege engines or elaborate fortifications. This measure also reflects having larger shares of the population not mainly engaged in primary production, proxied by the population of the largest settlement [71,72]. Further, more stratified and administratively complex societies–measured by the number of levels in administrative, military, and settlement hierarchies (combined here as one measure of hierarchical levels–see *SI*)–are hypothesized to be better equipped to implement useful technologies along with developing or adopting effective tactical and organizational models at scale. Thus, by this logic, increases in military technology should occur preferentially in larger scale societies. Previous analysis [53] reveals that these four dimensions are highly correlated within the Seshat sample and so represent an effective cross-cultural measure of societal scale to explore this hypothesis.

2. SocSoph ("social sophistication") represents the first PC of the Seshat variables *governance*, *infrastructure*, *information systems*, and *money*. This measure likewise derives from previous analysis of the dimensions of social complexity [53], capturing the important non-scale institutional and informational aspects. The hypothesis here is that societies with more sophisticated, pre-existing mechanisms for the exchange and implementation of ideas will generate and/or adopt innovations into widespread use at a faster pace.

3. Agricultural productivity (Agri) is the estimated yield of different regions, measured as tonnes per hectare of the major carbohydrate source consumed in each of our NGAs. These data are taken from the analyses in [73]. The term is included here to test the possibility that productivity affects the amount of resources that are available for technological advances.

Social scale and productivity, thus, give us two complementary views of the resource base that may drive the evolution of technology. Agri tracks the underlying material resource base in a given geographical region (our NGAs) needed to support development, including technological evolution. Social scale, on the other hand, is a measure of the territorial and population size of specific historical polities. Larger polities can gather resources from a large territory, including the human energy from large populations, even where agricultural productivity is

low. Separately SocSoph represents the sophistication of infrastructure and exchange media that could conceivably facilitate the flow of ideas from invention (whether within or outside of the society) to widespread adoption.

## Statistical analysis

In addition to standard correlational statistical analyses of our response and predictor variables, we used a general non-linear dynamic regression model to investigate factors affecting the evolution of military technology. This dynamic regression analysis distinguishes correlation from causation by estimating the influence potential causal factors at a previous time have on the response variable at a later time (known generally as Wiener-Granger causality [74,75]). While an improvement over 'static' correlations, where causal direction remains ambiguous, this method is, nevertheless, insufficient for making absolute claims of causality. Further scrutiny will be required to provide additional support for the provisional causal interpretations suggested below.

Our model takes the following form [70]:

$$Y_{i,t} = a + \sum_{\tau} b_{\tau} Y_{i,t-\tau} + c \sum_{i \neq j} \exp\left[-\frac{\delta_{i,j}}{d}\right] Y_{j,t-1} + h \sum_{i \neq j} w_{i,j} Y_{j,t-1} + \sum_{k} g_k X_{k,t-1} + \epsilon_{i,t}$$

Here $Y_{i,t}$ is the response variable (MilTech) for location (NGA) $i$ at time $t$. We construct a spatio-temporal series for Seshat response and predictor variables by following polities (or quasi-polities, such as archaeologically attested cultures) that occupied a specific NGA at each century mark during the sampled period. Thus, the time step in the analysis is 100 years.

On the right-hand side, $a$ is the regression constant (intercept). The next term captures the influences of past history ("autoregressive terms"), with $\tau$ = 1, 2, . . . indexing time-lagged values of $Y$ (as time is measured in centuries, $Y_{i,t-1}$ refers to the value of MilTech 100 years before $t$).

The third term represents potential effects resulting from geographic diffusion using our Space term. We used a negative-exponential form to relate the distance between location $i$ and location $j$, $\delta_{i,j}$, to the influence of $j$ on $i$. Unlike a linear kernel, the negative-exponential does not become negative at very large $\delta_{i,j}$, instead approaching 0 smoothly. The third term, thus, is a weighted average of the response variable values in the vicinity of location $i$ at the previous time step, with weights falling off to 0 as distance from $i$ increases. Parameter $d$ measures how steeply the influence falls with distance, and parameter $c$ is a regression coefficient measuring the importance of geographic diffusion. For an overview of potential effects resulting from geographic diffusion, see [69,76]; for a description of how we avoided the problem of endogeneity, see [70].

The fourth term detects autocorrelations due to any shared cultural history at location $i$ with other regions $j$ using our Phylogeny variable. Here $w$ represents the weight applied to the phylogenetic (linguistic) distance between locations (set to 1 if locations $i$ and $j$ share the same language, 0.5 if they are in the same linguistic genus, and 0.25 if they are in the same linguistic family). Linguistic genera and families were taken from *The World Atlas of Language Structures* and Glottolog [77].

The next term on the right-hand side represents the effects of the main predictor variables $X_k$ [70]; $g_k$ are regression coefficients. These variables (described in the previous section) are of primary interest because they enable us to test various theories about the evolution of MilTech against each other. Finally, $\varepsilon_{i,t}$ is the error term. We also include quadratic versions of these terms at a time lag (not shown) in order to explore non-linear responses to response and predictor factors.

Model selection (choosing which terms to include in the regression model) was accomplished by exhaustive search: regressing the response variable on all possible linear combinations of predictor variables. The degree of fit was quantified by the Akaike Information Criterion (AIC). Standard diagnostic tests were performed for the best-fitting models [70].

Missing values, estimate uncertainty, and expert disagreement in the predictors were dealt with by multiple imputation [78,79]. The response variable, MilTech, however, was not imputed as that can result in biased estimates [76]. For details of the multiple imputation procedure see *SI*. Because diagnostic tests indicated that the distribution of residuals are not gaussian, we used nonparametric bootstrap to estimate the *P*-values associated with various regression terms (see the *SI* for details). Additional robustness checks are similarly detailed in the *SI*.

## Results

### Spatio-temporal patterns

We first examined the frequency distributions of the variables of interest and the cross-correlations between WCs, overall MilTech, which combines all WCs, as well as calendar Time and the various aspects of social complexity and productivity. As expected, we find that all WCs are closely correlated with each other and with the overall MilTech variables. Plotting MilTech as a function of time for each NGA (Fig 4), we observe that there is a general upward trend, as expected. However, there are also periods when some technologies are lost, for a time. Most importantly, there is a great amount of variation between different geographic regions in the timing of MilTech increases. Interestingly, all WCs are more strongly correlated with the two dimensions of social complexity specified here–Scale and SocSoph–rather than with Time, suggesting that key drivers of MilTech evolution go beyond merely the additive nature of technology through the 'march of time'. The nature of any causality between complexity and MilTech is discussed below.

We next focus on "technology leaders", NGAs that at some point in their history had the highest value of MilTech available at the time. Fig 5 shows them, roughly in the order that they achieved world leadership (note that this order is also affected by how far back in the past we have data). The hot spot of technological development, through either innovation or adoption, appears to roughly coincide with the "Imperial Belt" of the Old World, located just south of the Great Eurasian Steppe (and in places, impinging into it, as in Sogdiana), which can be seen by the location of the 'leader' NGAs (mapped in Fig 5).

This same territory also of course corresponds roughly to the path of the overland silk routes used in our analyses (Fig 1). We return to this pattern below. Overall, the pattern is that most of the leading regions exhibit an increase in their overall MilTech levels roughly together and at a fairly regular, almost linear pace (after the 4$^{th}$ millennium BCE), with late comers accelerating at various points to merge with leaders. This is seen clearly in this graph on the example of Sogdiana, but it is a general pattern discernible in the regional examples (Fig 4).

We explored the "similarity" between NGAs by calculating the number of MilTech variables in each NGA shared with other NGAs at each time-step. As explained in *SI* (see *Similarity Analysis* and S1 Fig in S1 File) we trace how NGAs join the expanding Eurasian (eventually global) IN by noting the time when they achieve a similarity index of 10, that is, when they share 10 or more specific MilTech variables with one or more other NGAs. As the histograms in S1 Fig in S1 File show, the first NGAs that achieve this threshold of similarity appear between 3000 and 2500 BCE. As time progresses, more and more NGAs cross this threshold. Fig 6 maps the expansion of this IN–initially restricted to central Eurasia but growing eventually into a global network–by color coding the date when the NGA cross this threshold. Thus,

## (a) Europe and Africa

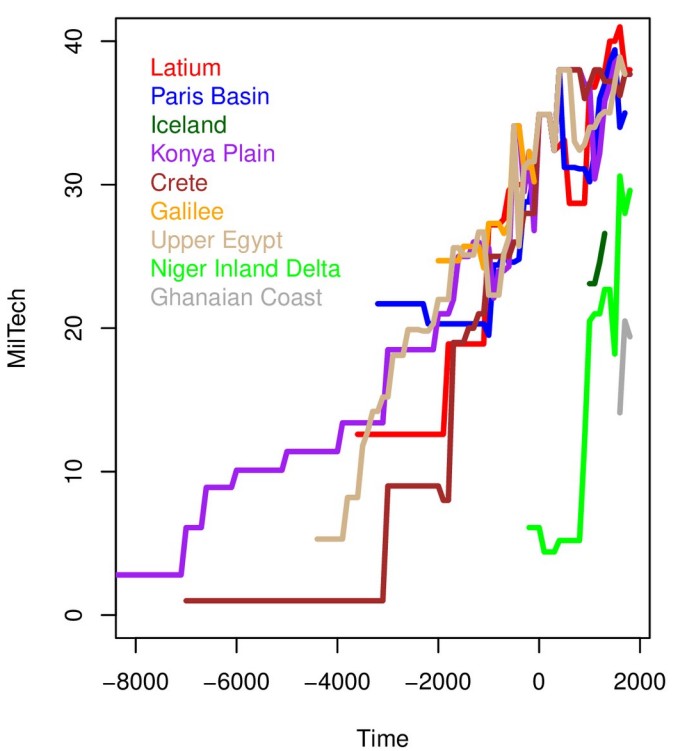

## (b) Western Asia

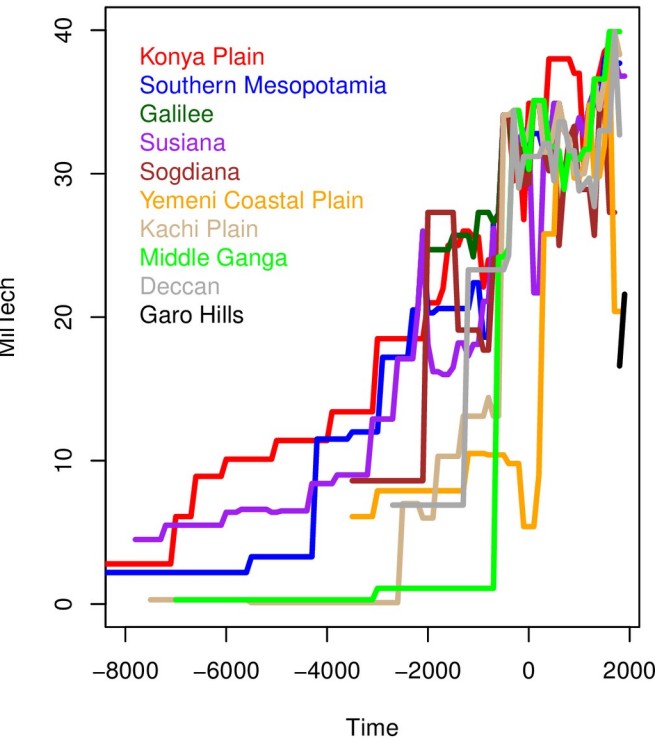

## (c) East and SE Asia

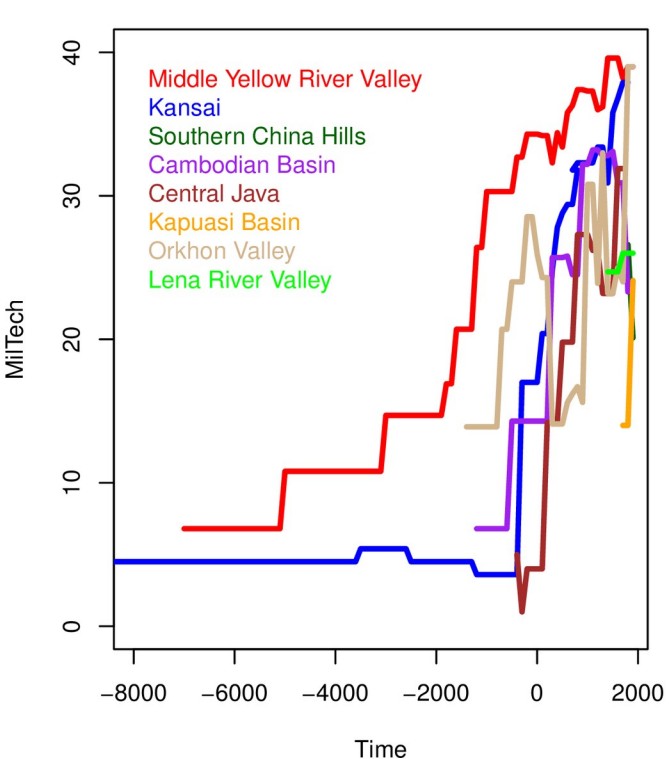

## (d) Americas and Oceania

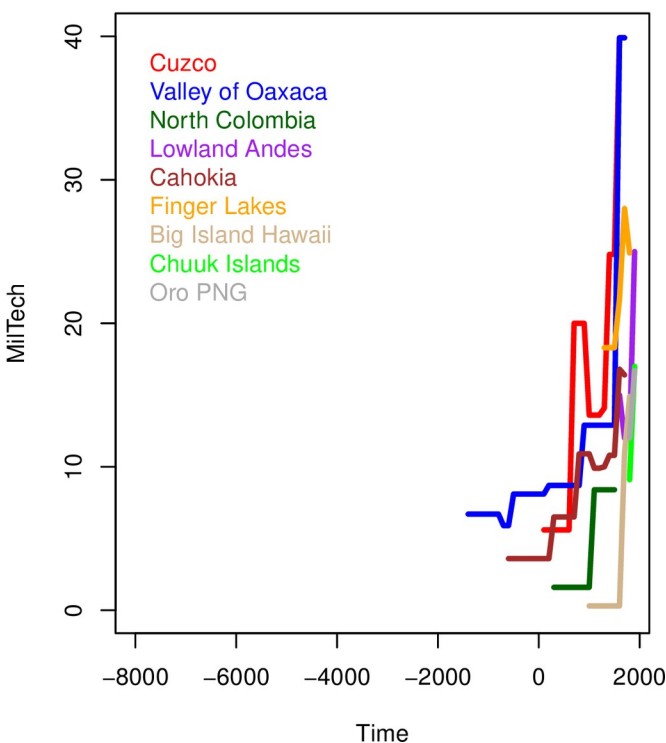

**Fig 4.** MilTech trajectories in Seshat NGAs, divided by major world region: (A) Europe and Africa; (B) Western Asia; (C) East and SE Asia; (D) Americas and Oceania.

the similarity analysis reveals that different regions not only saw rapid increases in their overall level of MilTech, but these areas came increasingly to share specific technologies. A plausible interpretation for this pattern is that, as the IN expands, each new region accelerates its development of MilTech to join the level achieved by the network leaders, until, eventually, all regions in diverse areas around the globe adopt similar 'MilTech packages'. Future work is needed to disentangle occasions where these late-comer regions adopt or adapt existing technologies from cases where 'leader' societies simply take over others, imposing their technologies (along with a host of other socio-political and cultural traits) onto this new regions.

**Dynamic regression results.** The best fitting model from our general dynamic regression analysis is shown in Table 1.

Our analysis identifies the following variables as having the strongest causal influence on MilTech:

- Autocatalytic effects (the value of MilTech in the previous time step).

- Global population size (WorldPop).

- Connection to an expanding (eventually global) Information Network (Centrality).

- Spread of Iron+Cavalry (IronCav), revealing both the importance of prior technology stock on continued technological evolution as well as the incentive that these advances placed on societies within connected information and competition spheres to adopt or develop additional technologies in response.

- Cultural similarity (Phylogeny), revealing that polities linguistically similar to polities with high MilTech are more likely to have high MilTech themselves. This effect could be a result of either common inheritance or easier diffusion of technology between culturally similar polities, or, most likely, both.

- Productivity of the resource base (Agri).

Investigation of the effects of Cavalry and Iron as predictor variables indicate that either, separately, has a statistically significant effect of similar strength on the evolution of MilTech. The synthetic variable, IronCav, however, is a better predictor than either of its constituents. For this reason, the results here are reported for IronCav only.

We estimated how location with respect to the system of Silk Routes affects the evolution of MilTech in each region. Our measure of Centrality (inverse distance to the nearest Silk Route node) finds strong empirical support, although we ran analyses using alternate methods of proxying this type of spatial effect (see *SI* for details). Overall our best model predicts the level of military technology with regression coefficient of determination ($R^2$) of 0.96. While some of this high predictability is a result of strong temporal autocorrelation, rerunning the regression omitting all autocorrelation terms nevertheless yields an $R^2$ of 0.72. Thus, more than 70% of the variation in MilTech is explained by WorldPop, Centrality, IronCav, and Agri.

We performed several supplemental analyses and robustness checks to detect any biases in our results. Several of these checks are discussed below and detailed in the *SI*.

Table 2 shows a comparison between the best fitting model and other models with $\Delta$AIC $\leq$ 2. Strong effects are detected in these alternative models for all terms in the best model including Agri, which, though its standardized coefficient is the smallest, remains statistically significant at the conventional $P < 0.05$ level. However, additional robustness tests

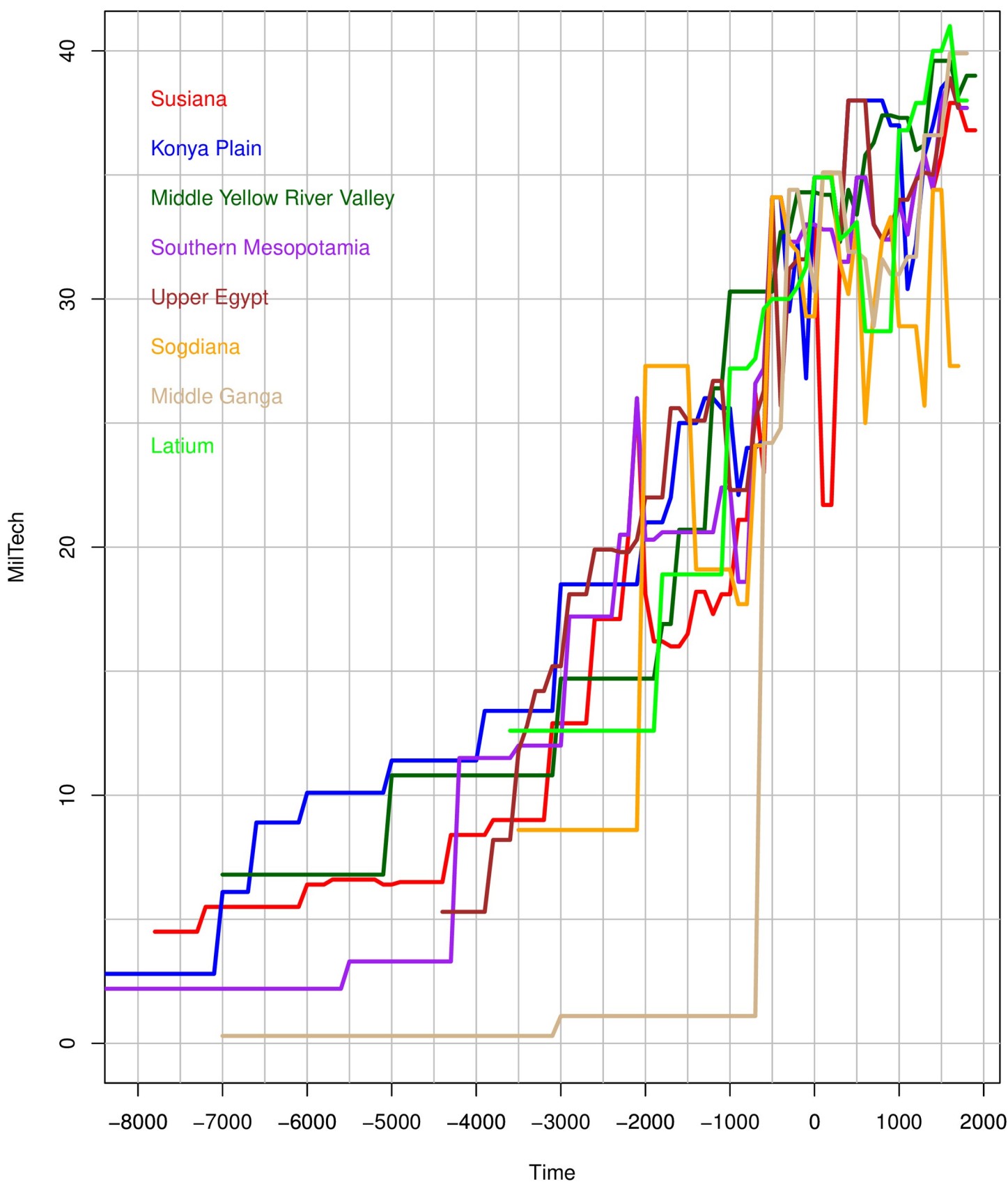

**Fig 5. "Technological leaders": NGAs that at some point achieved the highest MilTech score available at that time.**

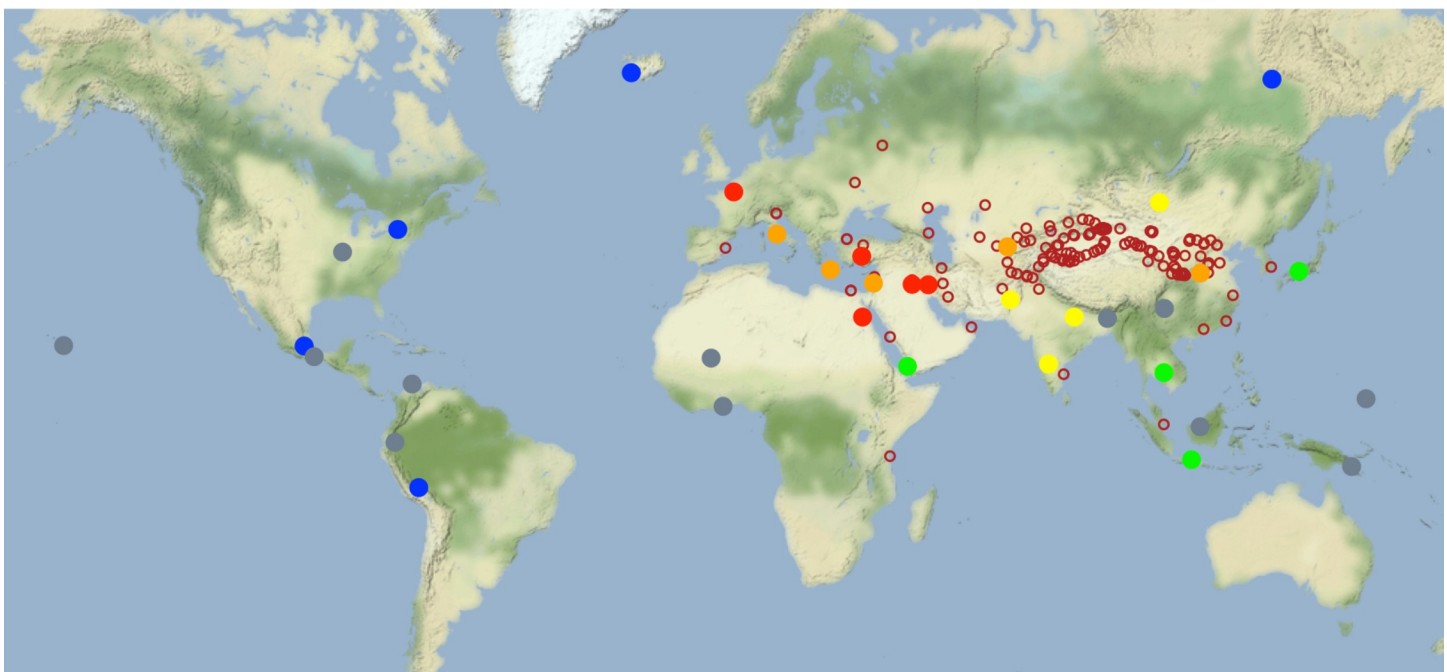

**Fig 6. Results of similarity analysis.** NGAs are binned into 6 categories according to the earliest time they share 10 or more MilTech variables with another NGA, displayed by color: dark red = 2500 BCE or before; orange = between 2500 and 1500 BCE; yellow = between 1500 and 500 BCE; green = between 500 BCE and 500 CE; blue = after 500 CE; grey = did not display any similarities during our sample period. Unfilled red circles indicate Silk Route Nodes as in Fig 3.

using multiple datasets built by random sampling from among the different WCs comprising the MilTech variable indicate that Agri is not always supported (see *SI* for details). Neither measures of social complexity, Scale and SocSoph, appear to have a consistent significant positive effect on MilTech evolution (they show up in several of the alternative models, but with small *t*-values and negative signs for Scale).

Further checks indicate that these results are robust to the inclusion of additional spatial and temporal autocorrelation effects: Neither geographic diffusion (Space) nor higher

**Table 1. Regression results for the best (lowest AIC) regression model.**

|  | Estimate | Std. Error | t value | Pr(>\|t\|) | Bootstrap estimated P |
|---|---|---|---|---|---|
| (Intercept) | 0.000 | 0.006 | 0.000 | 1.000000 | 0.521 |
| MilTech | 1.043 | 0.025 | 42.114 | < 2e-16 | 0.000 |
| MilTech.sq | -0.175 | 0.026 | -6.862 | 1.12e-11 | 0.000 |
| IronCav | 0.047 | 0.012 | 3.973 | 0.000076 | 0.000 |
| Agri | 0.020 | 0.008 | 2.542 | 0.011 | 0.028 |
| WorldPop | 0.039 | 0.011 | 3.505 | 0.00047 | 0.001 |
| Centrality | 0.027 | 0.008 | 3.375 | 0.00076 | 0.000 |
| Phylogeny | 0.037 | 0.008 | 4.486 | 8.01e-06 | 0.005 |

Estimate shows the standardized regression coefficients, which provide a direct measure of relative effects by the *lagged* predictors on the response variable. Thus, MilTech here represents the linear autoregressive term, AR(1). The column "t value" lists t-statistics, a measure of statistical significance of regression terms associated with various predictors. Pr(>\|t\|) is the statistical significance for regression assuming the Normal distribution of residuals, while Bootstrap estimated P is the result of nonparametric bootstrap that does not make this assumption.

**Table 2. Alternative model selection results.**

| MilTech | MilTech.sq | Scale | SocSoph | IronCav | Agri | WorldPop | Centrality | Phylogeny | ΔAIC |
|---------|-----------|-------|---------|---------|------|----------|------------|-----------|------|
| 42.11 | -6.86 | | | 3.97 | 2.54 | 3.50 | 3.38 | 4.49 | 0.00 |
| 41.83 | -6.70 | -0.61 | | 4.02 | 2.61 | 3.51 | 3.40 | 4.43 | 1.63 |
| 41.77 | -6.74 | | 0.20 | 3.92 | 2.38 | 3.51 | 3.37 | 4.47 | 1.96 |
| 41.74 | -6.73 | -0.98 | 0.79 | 3.97 | 2.43 | 3.57 | 3.42 | 4.32 | 3.00 |

The table shows t-statistics associated with each of the predictors (column headings) for the best models with ΔAIC (the AIC difference with respect to the best model) less than 2. The best model by AIC is included as the top row. An empty entry indicates that the term associated with this predictor is not included in the model.

temporal lags ($\tau$ = 2 centuries or greater) are significant. In addition, as we discussed in *Materials and Methods*, because our measure of MilTech includes the Metals and Animal WCs, which might confound the effect of IronCav due to a potential circularity, we re-ran the analysis using CoreMil, our measure of military technologies that does not include these WCs. This analysis yields essentially identical results (see *SI*), thus suggesting that the effect of IronCav is not spurious.

What is remarkable is that neither Scale nor SocSoph variables, which characterize polities, have any detectable effect on the level of MilTech. Overall, these results suggest that MilTech evolves almost entirely as an exogenous variable: it is little, if at all, affected by such polity characteristics as the population, territory size, the sophistication of information systems or administrative institutions, or provision of infrastructure and public goods.

As noted, our dynamic regression approach cannot offer a definitive demonstration that the factors in the best model are the central causal forces driving the evolution of military technologies. These variables may simply be highly correlated with the 'true' causal factors, not included in our analyses, or the causal link may be indirect, as these factors, as well as MilTech, could be caused separately by additional factors whose effects were felt at different time-scales. We explored such a possible 'hidden variable bias' as much as possible through supplemental analyses of several variables for which we had reliable information. As our findings remain robust to various tests, we provisionally conclude that these results offer appealing and parsimonious causal explanation for the long-run and global evolution of military technologies. Future research will need to scrutinize whether these results hold up to the inclusion of additional factors and exploration using alternate statistical methods or mechanistic models perhaps using agent-based modelling [24,27,80].

## Discussion

Our goals were to investigate the global spatio-temporal evolution of key pre-industrial military technologies to illuminate the major forces driving the evolution of these critical tools, whether by innovation, adoption and adaption, or a combination of these processes. Further, our approach to testing theoretically-informed hypotheses against a broad and diverse set of empirical historical data taken from *Seshat*: *Global History Databank* serves as an example of how more general patterns of technological evolution can be explored in future research, as well as more fine-grained analyses seeking to distinguish these different processes or explore the pathways taken by individual regions or societies. Here we surveyed various causal hypotheses, which together suggested that the evolution of military technology would be a function of some combination of global population size, connectedness to information exchange networks, involvement in inter-state competition networks, and prior histories of technological innovation and adoption (especially major breakthroughs such as iron metallurgy and horse riding), along with, perhaps, various properties of polities and their resource base. We set out

to test these theories empirically against the evidence from world history, using a stratified sample of polities in *Seshat*, dating from the Neolithic to the Industrial Revolutions.

While we found some empirical support for each of these hypotheses, no one theory alone accounted for the observed dynamics of military technology as well as a combination of the factors suggested by these various proposals. Our results not only explain why these theories have found support in previous studies, but also why a general understanding of the evolution of military technology has proven elusive. Our robust historical sample and extensive dynamic analyses allowed us to compare and combine elements of different theories proposed as critical drivers of military technology. Specifically, we found that global population size is a strong predictor of the subsequent levels of MilTech. While this result supports the Kremer-Taagepera model, it does not rule out other possible causal explanations based on additional variables, which, while correlated with global world population, could turn out to be a better predictor of MilTech. One such addition in future work could be to distinguish societies by their general affluence or social mobility [81], rather than treating populations as indistinguishable, which may play such a causal role driving both population increases and technological evolution.

Our analysis found that stock of prior technological innovations played an important role in the observed levels of military technology, not only from the autoregressive terms (again, supporting the Kremer-Taagepera model) but critically because the combination of iron metallurgy and horse riding had a particularly strong effect on innovation and adoption of militarily technologies in the periods under investigation here.

Importantly, we found that location within the central Eurasian IN was also a strong predictor of our response measure, in line with the insights of World Systems and cultural evolutionary theorists. This result supports the impact of being connected to other major centers of development and innovation, as well as being incorporated into spheres of inter-state competition.

However, it is noteworthy that geographic proximity between NGAs itself (proxied here by our Space measure) does not appear to be a strong predictor of the evolution of military technologies, contrary to what might be expected from certain cultural evolution theories and ideas of mimetic diffusion. This underscores the significance of iron and cavalry diffusion in particular, which have a strong effect on subsequent levels of MilTech, supporting previous work highlighting the unique role of the nomadic pastoralists of the Eurasian Steppe, early adopters of mounted archery tactics, in driving not only technological innovation among nearby agrarian populations, but in driving the expansion of social complexity and, relatedly, technological evolution throughout Afro-Eurasia [8,24,26,27,45–48,82]. The development of iron-smelting, as an input material for so many valuable weapons, appears to play a similarly crucial role [64–67]. These findings suggest that iron and cavalry were particularly critical technologies that conferred an important enough advantage that they fomented widespread adoption as well as sparked 'arms races' among competitors that included a host of other, related technologies as discussed above, which would explain the observed patterns.

This interpretation gains further support from our similarity analysis. Our main result indicate that the overall level of MilTech–measured with our aggregate MilTech score–generally rose over time (with some losses, noted above), with more and more regions coming to exhibit the same level of MilTech over time. Our similarity analysis unpacks this finding, demonstrating that not only did regions increasingly exhibit the same overall MilTech score, but they also came to share the same 'packages' of specific military technologies. Further, as expected, the regions with the highest combined similarity scores followed the same pattern as seen in the Centrality measure, as the NGAs closest to a Silk Route node both appeared as sharing MilTech variables with other NGAs earlier and continued to display similar MilTech packages

with other NGAs that joined the IN over time, resulting in their larger combined scores (see Fig 4).

An interesting and somewhat surprising finding is that the properties of polities, including such seemingly important characteristics as their scale (population and territory) and sophistication (e.g., information systems), have no significant impact on the evolution of military technologies wielded by the polity (with the partial exception of Phylogeny, discussed below). We expected both scale (Scale) and non-scale (SocSoph) aspects of social complexity to play a significant role in these processes, due to an increased availability of populations and resources to put towards technological development as well as how developments in organizational and informational-exchange capacities could facilitate the adoption and adaption of existing technologies from elsewhere. However, these terms display no significant effect on subsequent levels of MilTech, suggesting that the level of technology characterizing a particular polity (whether invented or adopted) depends not on the polity's characteristics, but rather on the characteristics of the inter-polity informational and competitive interaction spheres to which it belongs, along with the other factors identified above. The Arabian Peninsula, for example, despite being relatively low-scale in the early first millennium CE, adopts much of the 'military package' seen in other parts of Eurasia around 300 CE (see Fig 4 and the Similarity Analysis in the SI), as it became increasingly incorporated into Silk Route trade connections via the Persian and Roman imperial systems, before becoming its own seat of imperial power with the rise of Islam a few centuries later.

The only polity-related term that is included in the best regression model is Phylogeny, which can reflect an operation of one of two (or both) processes: inheritance of technological sophistication from a "common ancestor" (for example, Italy and France inheriting technologies from the Roman Empire), or easier spread of innovations between culturally similar countries (such as between Romance-speaking Italy and France, or between Arabic-speaking Egypt and Mesopotamia). The latter process likely reflects the greater likelihood that an innovation developed in one polity will be more compatible with existing institutional, social, cultural, and economic systems of a culturally similar polity than those of a more distant one [83]. One major component of this effect might be that military technologies require specific tactical and organizational apparatus to wield effectively. Cultural similarity then could not only facilitate exchange of information about a new, useful technology across societies, but facilitate the spread of knowledge of and increase the ability to acquire these more ephemeral aspects accompanying the material components of this new technology. Alternatively, linguistically similar polities might have engaged in more frequent and intense competition, which could lead to a similar impact (likely correlated strongly with the IronCav and Centrality effects) on overall MilTech. This is less plausible than the other processes, however, as interstate competition has been shown to be most intense involving culturally dissimilar polities [45,47,61]. Additional study is needed to fully clarify the different possible causal forces driving this effect and to explore the possible causal role that each of these potential processes play in the overall development and spread of these key military technologies, along with technological evolution more generally. Nevertheless, the finding that Phylogeny is a significant predictor of MilTech further speaks to the importance of connection-mediated information exchange, over and above closeness in space.

Lastly, we find that agricultural productivity, measured here as per-hectare tons of major carbohydrates, displays a significant effect on subsequent levels of MilTech. While we had no strong theoretical motivations for this idea, we included the term in analyses to test the possibility that an increased resource base would impact technological development. Its inclusion in our best model suggests that a certain level of agriculture productivity may have been a necessary component in generating and adopting new technologies. Perhaps a more efficient

productivity was required to support large enough populations not primarily employed in agriculture, or expanding a society's general resource base and extractive capacity provided the raw materials and intermediate goods used in constructing key military technologies. As noted, however, this factor displays a much weaker effect compared to the others, and is least robust to supplemental analyses. Thus, this result must remain tentative. Exploring more deeply the impact of agricultural productivity on the evolution of technology stands out as an important avenue for future research.

While these findings constitute an important first step towards identifying some of the major long-term drivers of technological evolution in general, and in the domain of military capacity in particular, and finding broad support for previously somewhat speculative theories, there is still much to be done to build on this line of research. First, it would be desirable to extend the geographical coverage beyond the stratified global sample used in the present study, particularly relating to the phylogenic connections in the spread of existing technologies and the different possible processes that lead to this interesting effect. Second, it would be important to explore the downstream consequences of changes in military technology for other aspects of human life, including levels of peacefulness (or, alternatively, mortality rates due to violence), equality (e.g. distributions of wealth, rights of citizenry, levels of exploitation and oppression based on class or ethnicity) and public health (e.g. longevity, infant mortality, nutrition, infection rates, etc.). Third, our goal was to offer a preliminary exploration of some key causal forces proposed to support the evolution of military technology, ignoring differences between the initial innovation of new technologies and subsequent adoption by other societies. Future work is needed to pinpoint the source(s) of invention and distinguish advances made by innovation from advances by later spread to assess whether the same or different factors drive each of these separate processes. Fourth, additional potential drivers of technological innovation in general should be explored, over and above the effects of population size, connectivity, and existing stocks of critical innovations, as well as analyzing further the potential causal role played by rising agricultural productivity. These explorations would include factors impacting resource scarcity (e.g. due to drought, pestilence, and other natural disasters), more direct measures of intergroup competition (e.g. levels and intensities of external warfare, cultural distance between competitors, and other exogenous factors), identifying various different regional INs which might (partially) overlap in time and space.

Finally, it is important for future studies to 'narrow in' on the details of some of the more macro-level processes suggested by the present study. In particular, it will useful to explore the possible impact of regional-level factors along with a broader range of technological innovations within the polity (e.g. in energy, construction, transportation, and information sectors). Seshat data is relatively coarse, resolved here to 100-year intervals. While this granularity is well suited to exploring broad, global-level dynamics over thousands of years, it likely misses some of the nuances and outlying patterns. Future effort can hopefully generate more fine-grained temporal data allowing for meso- and even micro-level scrutiny of the pathways to technological evolution taken by different societies in various times and places. Alongside this, we require more qualitative investigation into the details of the specific items as well as the less material, tactical and managerial aspects of technological development employed in a host of specific historical cases.

Beyond the insights gained from these analyses on the development of military technologies over the very long-term, we hope that the approach presented here, which explores likely casual theories against a wide body of empirical data gathered by the *Seshat* project, will provide a roadmap to these important future studies, allowing scholars to delve deeper into not only the critical 'Military Revolutions' throughout history, but into the evolution of technology generally.

## Supporting information

**S1 File. Supporting information text and figs.**
(DOCX)

**S1 Data. Compressed file containing data files and analysis scripts.**
(ZIP)

## Acknowledgments

The authors are grateful to Sergey Nefedov who reviewed data and provided helpful comments. We thank also Christopher Chase-Dunn, Peter Grimes, Gene Anderson, and the *SetPol* project for their constructive critique on earlier versions of the manuscript, as well as Jennifer Larson and Alan Covey for helpful comments on previous drafts. We gratefully acknowledge the contributions of our team of research assistants, post-doctoral researchers, consultants, and experts. Additionally, we have received invaluable assistance from our collaborators. Please see the Seshat website (www.seshatdatabank.info) for a comprehensive list of private donors, partners, experts, and consultants and their respective areas of expertise.

## Author Contributions

**Conceptualization:** Peter Turchin, Daniel Hoyer, Andrey Korotayev, Nikolay Kradin, Sergey Nefedov, Gary Feinman, James S. Bennett, Pieter Francois, Harvey Whitehouse.

**Data curation:** Peter Turchin, Daniel Hoyer, Jill Levine, Jenny Reddish, Enrico Cioni, Chelsea Thorpe, Pieter Francois, Harvey Whitehouse.

**Formal analysis:** Peter Turchin, James S. Bennett.

**Funding acquisition:** Peter Turchin.

**Investigation:** Daniel Hoyer, Andrey Korotayev, Nikolay Kradin, Sergey Nefedov, Gary Feinman, Jill Levine, Jenny Reddish, Enrico Cioni, Chelsea Thorpe, James S. Bennett, Pieter Francois, Harvey Whitehouse.

**Methodology:** Peter Turchin, Daniel Hoyer, Andrey Korotayev, Nikolay Kradin, Sergey Nefedov, Gary Feinman, Pieter Francois, Harvey Whitehouse.

**Project administration:** Peter Turchin, Daniel Hoyer, Pieter Francois, Harvey Whitehouse.

**Supervision:** Pieter Francois, Harvey Whitehouse.

**Visualization:** Peter Turchin, Daniel Hoyer, James S. Bennett.

**Writing – original draft:** Peter Turchin, Daniel Hoyer.

**Writing – review & editing:** Peter Turchin, Daniel Hoyer, Andrey Korotayev, Nikolay Kradin, Sergey Nefedov, Gary Feinman, Jill Levine, Jenny Reddish, Enrico Cioni, Chelsea Thorpe, James S. Bennett, Pieter Francois, Harvey Whitehouse.

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
