## [Decision Letter · Decision Letter 0]

25 Jun 2021

PONE-D-21-17737

Rise of the War Machines: Charting the Evolution of Military Technologies from the Neolithic to the Industrial Revolution

PLOS ONE

Dear Daniel Hoyer,

I have received two reviewers' report on your manuscript and decided I would not wait for the third reviewer that I had invited. The two reviewers, I'm happy to say, concur that your manuscript is an important and valuable contribution to the cultural evolution literature. I share their point of view. Each reviewer offers specific recommendations that I would ask you to follow closely when revising the paper.

Reviewer 1 remarks that reducing the history of military technology to "hard" technologies (e.g. weapons or transportations) neglects important aspects of war tactics: "military doctrines, logistics, tactical skill, organizational structures, and ability to learn". In other words, your study captures the evolution of tactical "hardware", but not that of tactical "software", so to speak. They note that a more qualitative approach could capture this dimension. Please discuss this possible limitation of your study in the revision. Reviewer 1 also asks you to specify how your analysis can distinguish between technological change due to innovation vs. diffusion.

Reviewer 2 makes two specific and useful recommendations regarding your statistical analysis. The first is to make sure that the way you bundle military-technology related variables is justified. In other words, you should rule out the possibility that the effects you document are driven by one or a few outlier variables. Please implement the analysis they recommend, i.e., analysing random subsets of the 46 variables. Re-running your models with a subset of military technology variables is something you already do when you substitute "Core MilTech" for "MilTech". Reviewer 1 would like you to generalise this method to multiple, randomly selected subsamples. Reviewer 2 also suggests an alternative analysis for the impact of existing stock of technologies on the progress of military technologies.

In addition to these remarks, I have a number of editorial recommendations of my own, detailed below. Provided the revision appears to fulfil the reviewers' requirements and mine, I may not be sending it back for a new round of reviews.

My recommendations are either stylistic or related to the way statistical results are reported.

Concerning the reporting of results, my main concern is related to the repeated causality claims made in the manuscript. Given the data and methods, the results establish (at best) predictive causality in the sense of Granger. This type of causality cannot be equated with causality *simpliciter*, among other things because it does not rule our latent confounding effects. Please qualify all claims related to causality or causal inference.

I suspect there is a significant mistake in Table 1, which lists both MilTech and MilTech sq. as variables in the best-fitting model. This seems to contradict the description given on l. 301, but more importantly including two versions of the same variable in the same model raises obvious issues (multicollinearity, etc.). I suspect that you in fact tested 2 versions of the model, one with MilTech and the other with MilTech sq.

In general the manuscript in its current version assumes too much familiarity with the Seshat database and project, its organisation and its acronyms. Key constructs like "NGA" or "IN" are not explained or even glossed. Other constructs, like the "Scale" variable, will be familiar to readers who already read Seshat-based studies, but are only cursorily explained here (the characterisation given on l. 279 is insufficiently clear). For yet other variables like "SPC1" an explanation is promised but never provided (l. 349). There is also a tendency to use your own abbreviations for standard statistical constructs (e.g. using "delAIC" for ∆AIC or delta AIC). What is missing is not only a basic explanation of the Seshat lingo, but a sense of why the constructs were designed in the way that they are — for instance, why the Scale variable is a better measure of a polity's size than other possible measures.

We look forward to receiving your revised manuscript.

Kind regards,

Olivier Morin

Academic Editor

PLOS ONE

Journal Requirements:

We note that one or more of the authors are employed by a commercial company: "Complexity Science Hub."

Reviewers' comments:

Reviewer's Responses to Questions

**Comments to the Author**

1. Is the manuscript technically sound, and do the data support the conclusions?

Reviewer #1: Partly

Reviewer #2: Yes

2. Has the statistical analysis been performed appropriately and rigorously? 

Reviewer #1: Yes

Reviewer #2: Yes

3. Have the authors made all data underlying the findings in their manuscript fully available?

Reviewer #1: Yes

Reviewer #2: Yes

4. Is the manuscript presented in an intelligible fashion and written in standard English?

Reviewer #1: Yes

Reviewer #2: Yes

5. Review Comments to the Author

Reviewer #1: This is a very valuable paper providing the most systematic large-n analysis to date of the evolution of military technology. It should acknowledge more clearly that large-n studies are just one of the tools needed for a proper explanation, but should definitely be published.

(Editor's note: I am here  pasting Reviewer 1's attached comments since they may not be automatically added to this action letter for word limit reasons. OM)

This is a valuable paper, making a real contribution to a significant debate. My one reservation is that I think this kind of large-n survey usually has to be combined with more detailed work to explain something as complicated as military technologies. Military historians typically find that technology is only one part of effectiveness, and not always the most important part. The most interesting result in this paper is the lack of fit between SocSoph and technological gains (lines 455-56, 532-36), but I wonder how much of this is because technologies are little use without appropriate doctrine (the central point in Stephen Biddle’s book Military Power [2004]). Fig. 5 illustrates this—looking at the scores for Latium around 2kya, they’re not very different from other societies, because Roman military technology wasn’t actually that different from other Mediterranean, Middle Eastern, and European societies (as measured in the categories at lines 287-98 in the supporting material); however, the Roman army’s ability to apply these technologies transformed what they meant in practical terms, and that was probably determined primarily by Roman SocSoph. I also immediately think of the classical Greeks, whose military technologies were very ordinary but their application of them was extraordinary.

One difficulty for large-n surveys is that while technological categories are relatively easy to identify in the archaeological record, military doctrines, logistics, tactical skill, organizational structures, and ability to learn aren’t; and even when ancient and medieval writers describe such doctrines, there’s no obvious way to score them. This paper should absolutely be published, because there’ve been few (if any) large-n surveys of this level of sophistication, but the main point it illustrates is perhaps that large-n surveys are only the beginning of the study of warfare, and we have to follow up on their results with analytic narratives and case studies.

I also thought that the paper should distinguish more between technological innovation and diffusion. I’m currently reading two case studies of the Comanches, who probably had the best light cavalry in the world in the 18th century. They didn’t invent light cavalry, because there were no horses in North America to invent it with before the Spaniards brought them, but they did adopt and then adapt horse-herding and perfect styles of mounted fighting much better than the Apaches or even the Spaniards and Mexicans. They didn’t adopt or adapt firearms because their light-cavalry tactics made muzzle-loading muskets irrelevant. The Comanches were eventually defeated because Texans adopted and adapted Comanche tactics in the 1830s and combined them with revolvers and, later, breech-loading rifles that had been invented on the US East Coast. Neither the Comanche nor the Texans could have invented revolvers and rifles themselves, but the US Army failed to figure out how to use the new technologies until Texans put them together with tactics in new ways; but the final Comanche defeat in the 1870s depended on the US Army learning from the Texans and then exploiting the scale of US federal infrastructure. 

These points aren’t just details that can be subsumed within a large-n model; we’re missing what really happened if we only see the coarse-grained technological/geographical narrative. The paper should be clearer that while a large-n survey is a necessary condition for a good explanation, it’s not sufficient, and should be treated as a starting point for other kinds of analysis.

One final detail: the throwaway comment on naval warfare on line 69 isn’t adequate. 

Reviewer #2: This paper examines the long-term evolution of military technologies using the Seshat: Global History Databank. Creating a series of composite variables, which use simpler variables as a proxy for more complex ones such as Information Complexity or Military Technologies, the authors then test several theoretical claims in the literature. Several factors seem to significantly predict the advancement of Military Technology (MilTech), e.g., global population size, cultural similarity, and the spread of iron and cavalry. They also find variables that do not appear to dramatically influence of the level of military technology. Specifically, with the notable exception of phylogeny, characteristics such as the scale and sophistication of a polity appear to be non-significant predictors. This leads them to conclude that military technology evolves, for the most part, as an exogenous variable.

Overall, I applaud the authors for doing a very thorough and ambitious investigation in the evolution of military technology. I believe the paper will provide a catalyst for many follow up studies and serve a vital role of stimulating scientific debate. I do, however, have two general points that I’d like to make.

For my first point, I want to highlight a potential limitation of this approach, in that aggregation might mask a simpler explanation for the results of your model. That is, your predictors might be predicting only a subset of your 46 binary variables used to create your MilTech measure. One way to deal with this is to create alternative MilTech measures by randomly sampling from your set of 46 variables. What you do here is create multiple MilTech measures composed from different, randomly sampled subsets of your 46 variables. How much does this change your overall results in this study? It might be the case that what your model is predicting is a specific subset of technologies that disproportionately influence the overall result. If the result is not robust to these random subsets, then I believe it diminishes your claim that the variables are predicting military technology per se. Instead, it might be the case that your models are predicting a specific subset of the technologies, as opposed to an aggregate measure of overall military technology.

My second point relates to the variable for the existing stock of technologies. Here, you use the existing stock of technology as a proxy for the influence of current technological level on military technological evolution. This is done in two ways: as a temporal autoregressive process and focusing on horse riding and iron smelting. For the temporal autoregressive measure, you could also look at a related approach such as transfer entropy. The advantage of this is that it does not rely on using a single timeseries to predict its future state. Instead, you can use separate timeseries (X and Y), as predictors of one another. This tells you how much uncertainty is reduced in the future values of Y by knowing the past values of X given the past values of Y. It is a way of measuring the influence of one timeseries process (X) on another timeseries process (Y). I feel that this would help you disentangle the directionality of the relationship between some of your variables (as the measure is non-symmetric and X|Y does not equal Y|X). So, for instance, you could have the existing stock of technology as one timeseries and the miltech variable as another timeseries. You would predict that the information flow goes from the existing stock to miltech, but not necessarily the other way around.

Finally, a minor point is that for the results in Table 1, you should probably use scientific notation for extremely small p-values (especially for the MilTech and MilTech.sq p-values where you just have 0). In fact, after writing this comment, I noticed you already did this in your supplementary materials (so it should be easy to address).

6. PLOS authors have the option to publish the peer review history of their article (what does this mean?). If published, this will include your full peer review and any attached files.

Reviewer #1: No

Reviewer #2: No

---

## [Author Response · Author response to Decision Letter 0]

7 Aug 2021

Our response to Reviewer and Editor comments are included in the uploaded file 'Response to Reviewers.docx'. We copy our responses below as well

Response to Editor

We thank both peer reviewers for their thoughtful and helpful comments as well as the Editor’s helpful summary and additional comments, and for allowing us the chance to revise our manuscript. Both reviewers had very favorable things to say about our analytical approach and description of results, each concluding that the paper will make a valuable contribution to the field. 

The main concerns seemed to revolve around ensuring that we properly motivate our analytical approach and test the robustness of results to alternate methods, as well as properly qualifying our interpretations in light of the limitations of our methods. This later point is in line with the Editor’s main concern about our overly-ambitious causal inferences. We feel these are valid and quite useful critiques. We have made numerous edits and performed additional supplementary analyses to clarify our approach and interpretations and provide fuller support for our claims, while also highlighting the need for future work and alternative methods to further explore the evolution of military technology. Overall, our manuscript is much improved as a result of these comments, and we are grateful to the Reviewers for pointing out shortcomings in how we described and justified some of our central arguments. 

Lastly, in the response to the original submission the PLOS editors requested that we add a Competing Interests Statement and amend the Funding Statement based on the idea that the Complexity Science Hub, Vienna (CSH), with which some of our coauthors are affiliated, is a commercial company. This is incorrect, as the CSH is a non-profit research institute based in Vienna. It is registered as a Verein (‘public association’) with the Austrian Vereinsbehoerden (you can find it listed as well with the Research Organization Registry, id: ror.org/023dz9m50). We have attached a file from the Austrian government regarding the CSH’s status. If you require further information about the CSH and its non-profit status, please contact Philipp Marxgut (marxgut@csh.ac.at), CSH Secretary General. I have reaffirmed the lack of competing interests with all coauthors.

Responses to Editor Comments

Reviewer 1 remarks that reducing the history of military technology to "hard" technologies (e.g. weapons or transportations) neglects important aspects of war tactics: "military doctrines, logistics, tactical skill, organizational structures, and ability to learn". In other words, your study captures the evolution of tactical "hardware", but not that of tactical "software", so to speak. They note that a more qualitative approach could capture this dimension. Please discuss this possible limitation of your study in the revision. Reviewer 1 also asks you to specify how your analysis can distinguish between technological change due to innovation vs. diffusion.

See below for responses to R1’s comments

Reviewer 2 makes two specific and useful recommendations regarding your statistical analysis. The first is to make sure that the way you bundle military-technology related variables is justified. In other words, you should rule out the possibility that the effects you document are driven by one or a few outlier variables. Please implement the analysis they recommend, i.e., analysing random subsets of the 46 variables. Re-running your models with a subset of military technology variables is something you already do when you substitute "Core MilTech" for "MilTech". Reviewer 1 would like you to generalise this method to multiple, randomly selected subsamples. 

This is a very good suggestion. We implemented this by using a kind of bootstrap, generating multiple response variables by randomly resampling Warfare Components that make up our primary MilTech measure. We repeated the procedure 100 times to obtain and analyze the 100 best models by AIC. This analysis revealed that the most strongly supported terms found in the primary analysis remain supported, appearing in the best models in all or all-but-one instance. The only significant difference is the Agri term, which appeared in only 64 of the 100 cases using this bootstrap method. 

This analysis suggests that randomly choosing WCs has a small effect on our results, with the exception of Agri. The implication is that different WCs can substitute for each other with little loss of information, something that was already hinted at with the comparison between MilTech and Core MilTech.

We have included this supplementary analysis in the SI (under the subheading ‘Testing bias in aggregation of MilTech variables’) with a description of results. We also added a discussion of this robustness check in the main text (in the Dynamic Regression Results section) and a paragraph in the Discussion detailing the more limited support that Agri receives as a key driver of MilTech compared to the other factors. 

Reviewer 2 also suggests an alternative analysis for the impact of existing stock of technologies on the progress of military technologies.

See our response below.

In addition to these remarks, I have a number of editorial recommendations of my own, detailed below. Provided the revision appears to fulfil the reviewers' requirements and mine, I may not be sending it back for a new round of reviews.

My recommendations are either stylistic or related to the way statistical results are reported.

Concerning the reporting of results, my main concern is related to the repeated causality claims made in the manuscript. Given the data and methods, the results establish (at best) predictive causality in the sense of Granger. This type of causality cannot be equated with causality simpliciter, among other things because it does not rule our latent confounding effects. Please qualify all claims related to causality or causal inference.

This is a very good point and we agree that our original language did not adequately convey the appropriate qualifications to our interpretation of results. We have edited text throughout the MS to make this more clear. 

Specifically, we added an explanation in the Statistical Analysis section explaining how we use the dynamic analyses to support Granger-type causal inference, but stressing also that these inferences are partial and provisional and need further study to support. We have also added text at the end of the Results section clarifying that our interpretations are based on the predictive relationships between our (theoretically-motivated) predictor and response variables. We stress as well that our interpretations are provisional and subject to additional analyses and alternate approaches. Finally, we have modified text throughout the Discussion section to qualify the conclusiveness of our claims and highlight the suggestive nature of these results, requiring future work to explore further.

I suspect there is a significant mistake in Table 1, which lists both MilTech and MilTech sq. as variables in the best-fitting model. This seems to contradict the description given on l. 301, but more importantly including two versions of the same variable in the same model raises obvious issues (multicollinearity, etc.). I suspect that you in fact tested 2 versions of the model, one with MilTech and the other with MilTech sq.

This is not a mistake. We routinely perform checks for nonlinearity in how predictors affect the response variable. This check showed that the relationship between MilTech(t+1) and MilTech(t) (that is, lagged value of the response variable) is curvilinear. We capture this nonlinearity with a quadratic term, MilTech (t) squared. There is a strong statistical support for this form, because both the linear and quadratic MilTech have high t-values, and P << 0.0001. For this reason, we retain both terms in the best-fitting model. The interpretation of this statistical result is that the evolution of MilTech is characterized by stabilizing selection, with the equilibrium trajectory set by the predictors. If a random perturbation moves MilTech level away from the equilibrium, the negative quadratic term ensures that MilTech returns to it. 

We have added text in the Statistical Analysis section clarifying the inclusion of this square term. Detailing the statistical model, we now note: "We also include quadratic versions of these terms at a time lag (not shown) in order to explore non-linear responses to response and predictor factors. "

In general the manuscript in its current version assumes too much familiarity with the Seshat database and project, its organisation and its acronyms. Key constructs like "NGA" or "IN" are not explained or even glossed. Other constructs, like the "Scale" variable, will be familiar to readers who already read Seshat-based studies, but are only cursorily explained here (the characterisation given on l. 279 is insufficiently clear). For yet other variables like "SPC1" an explanation is promised but never provided (l. 349). There is also a tendency to use your own abbreviations for standard statistical constructs (e.g. using "delAIC" for ∆AIC or delta AIC). What is missing is not only a basic explanation of the Seshat lingo, but a sense of why the constructs were designed in the way that they are — for instance, why the Scale variable is a better measure of a polity's size than other possible measures.

We are grateful for bringing to our attention this deficiency in our descriptions. We have modified text throughout to provide additional details about our methods and terms used as part of the Seshat project. Firstly, we explain our data coding procedure and define key terms early on in the main text (in A General Approach to Quantifying the Evolution of Pre-Industrial Societies subsection), and have added more detail to the descriptions of the factors used in analyses in Materials and Methods (‘Hypotheses to be Tested: Defining Predictor Variables’ subsection), particularly the Scale and SocSoph factors. We also clarified our NGA-based sampling procedure and edited text in the SI to clarify what the different factors were (e.g. the relationship between Scale and SocSoph to each other and as different dimensions of social complexity, as well as further detail on how and why we constructed the alternate response measure CoreMil). 

Responses to Reviewer 1 Comments

This is a valuable paper, making a real contribution to a significant debate. My one reservation is that I think this kind of large-n survey usually has to be combined with more detailed work to explain something as complicated as military technologies. Military historians typically find that technology is only one part of effectiveness, and not always the most important part. The most interesting result in this paper is the lack of fit between SocSoph and technological gains (lines 455-56, 532-36), but I wonder how much of this is because technologies are little use without appropriate doctrine (the central point in Stephen Biddle’s book Military Power [2004]). Fig. 5 illustrates this—looking at the scores for Latium around 2kya, they’re not very different from other societies, because Roman military technology wasn’t actually that different from other Mediterranean, Middle Eastern, and European societies (as measured in the categories at lines 287-98 in the supporting material); however, the Roman army’s ability to apply these technologies transformed what they meant in practical terms, and that was probably determined primarily by Roman SocSoph. I also immediately think of the classical Greeks, whose military technologies were very ordinary but their application of them was extraordinary. One difficulty for large-n surveys is that while technological categories are relatively easy to identify in the archaeological record, military doctrines, logistics, tactical skill, organizational structures, and ability to learn aren’t; and even when ancient and medieval writers describe such doctrines, there’s no obvious way to score them. This paper should absolutely be published, because there’ve been few (if any) large-n surveys of this level of sophistication, but the main point it illustrates is perhaps that large-n surveys are only the beginning of the study of warfare, and we have to follow up on their results with analytic narratives and case studies.

This is an excellent point and we agree that the material components of technological development are only one part of the story. The tactical, organizational, and managerial aspects that go into how these technologies are wielded certainly play key roles in their effect. Here we are primarily interested in the evolution of the technologies themselves rather than the other aspects, both because, as R1 notes, coding these more ephemeral aspects is very difficult (though certainly possible using our approach to similar doctrinal ideological categories; e.g. Mullins et al. 2018, Whitehouse et al. forthcoming) and beyond the scope of what we could include here, but also because part of the motivation for the present study is to model an approach that could be extended to such follow-up questions. 

We have added text in various places in the main text to make this explicit. In particular, we added to the Discussion section text noting the limitations of our approach and the relatively coarse detail of our data, remarking how more fine-grained study and qualitative investigation are important areas for future research. 

It is worth noting too that we entirely agree that the lack of significant effect seen between SocSoph and our response measure is an important (and somewhat surprising) result. It may very well have been expected that the SocSoph measure would mediate the ability of societies to effectively wield technologies as R1 notes; namely, that societies with more sophisticated information storage and exchange systems would be better equipped to develop or adapt the tactical and organizational mechanisms that accompany material technologies. Yet, as noted we do not see a significant relationship between SocSoph and MilTech or any other polity-specific factors. As we discuss in the main text, we interpret this as suggesting that MilTech is driven largely by extra-polity processes. The accompanying tactics and other ephemeral aspects that accompany technological change may develop after the technologies themselves have been widely adopted, or evolve through separate processes; again, future work exploring tactical and organizational aspects of technological evolution would be needed to provide insight into this open question, which is beyond the scope of the present study but we feel is ripe for investigation using the Seshat approach evidenced here; we have added text at the end of the Discussion section remarking on this. 

Finally, the tactical knowledge aspect of technological change may be reflected to some degree in our measure of cultural similarity (Phylogeny). Indeed, it may be that the tactical knowledge accompanying major technological developments (or standing on their own merits) is driven by external competition or other forces, and is then spread more effectively to culturally similar societies than dissimilar ones, since closeness in both language and socio-cultural systems could facilitate the adoption and adaption of these tactics. This may help to explain the result showing that Phylogeny is a significant predictor of subsequent MilTech, in addition to the benefits of exchanging information about the material and mechanical components of new technologies. We have added discussions to this effect in the Discussion section where we treat this finding. We feel that our overall findings are strengthened by this additional dimension and thank R1 for raising this very important and interesting aspect of our results that we had not fully considered before.

Ref:

Mullins, Daniel Austin, Daniel Hoyer, Christina Collins, Thomas Currie, Kevin Feeney, Pieter François, Patrick E. Savage, Harvey Whitehouse, and Peter Turchin. “A Systematic Assessment of ‘Axial Age’ Proposals Using Global Comparative Historical Evidence.” American Sociological Review 83, no. 3 (May 8, 2018): 596–626. https://doi.org/10.1177/0003122418772567. 

Whitehouse, Harvey, Pieter François, Daniel Hoyer, Kevin Chekov Feeney, Enrico Cioni, Rosalind Purcell, Robert M. Ross, et al. “Big Gods Did Not Drive the Rise of Big Societies throughout World History.” OSF Preprints, April 1, 2021. https://doi.org/10.31219/osf.io/mbnvg.

I also thought that the paper should distinguish more between technological innovation and diffusion. I’m currently reading two case studies of the Comanches, who probably had the best light cavalry in the world in the 18th century. They didn’t invent light cavalry, because there were no horses in North America to invent it with before the Spaniards brought them, but they did adopt and then adapt horse-herding and perfect styles of mounted fighting much better than the Apaches or even the Spaniards and Mexicans. They didn’t adopt or adapt firearms because their light-cavalry tactics made muzzle-loading muskets irrelevant. The Comanches were eventually defeated because Texans adopted and adapted Comanche tactics in the 1830s and combined them with revolvers and, later, breech-loading rifles that had been invented on the US East Coast. Neither the Comanche nor the Texans could have invented revolvers and rifles themselves, but the US Army failed to figure out how to use the new technologies until Texans put them together with tactics in new ways; but the final Comanche defeat in the 1870s depended on the US Army learning from the Texans and then exploiting the scale of US federal infrastructure. 

We entirely agree that the distinction between invention and later diffusion is a critical one to make to understand the histories of technological spread and use. However, as noted above we intend for this piece to be only the opening salvo in a line of work investigating the various aspects of technological evolution, military and more generally. We feel it is important to lay the foundation and model an approach to these such future work by focusing on technological evolution more broadly – which, as we define here, includes innovation as well as adoption and adaptation. This is why we conflate these different processes in this piece. 

We have added text to make this more clear to readers. Specifically, in the Introduction we clarify how we define ‘technological evolution’ for the purposes of this study. We also clarify in the Results and Discussion sections that change in MilTech can occur through multiple mechanisms, e.g. when describing regional patterns in the pace of technological change (distinguishing early-adopters from late-joiners to the growing global information network) we explain that we do not distinguish innovation from later adoption. We also highlight that disentangling these process will be an important focus of future studies. 

The example R1 cites is very interesting. From the perspective of the evolution of military technology broadly, this case actually illustrates well the processes identified by our analyses; various technologies (and tactics) not invented in North America are introduced as the area becomes entangled in the information exchange networks of western Europe (through conquest, settlement, and expansion) and then these become widespread in use within a context of fierce interstate competition. While our coarse-grained macro-level model can not predict which group would win out in this competition, the key point from this level of analysis is that information exchange and inter-group competition seem to have facilitated the spread of key military technologies (herding, mounted archery, hand-held firearms) into an area that did not have them previously

These points aren’t just details that can be subsumed within a large-n model; we’re missing what really happened if we only see the coarse-grained technological/geographical narrative. The paper should be clearer that while a large-n survey is a necessary condition for a good explanation, it’s not sufficient, and should be treated as a starting point for other kinds of analysis.

We, indeed, agree that more fine-grained and detailed study is a necessary component of reconstructing social dynamics, as noted above. We would argue though that neither the fine-grained nor the broader, macro-level approach reveals what ‘really happened’, rather that both are needed together to untangle these complicated patterns; these are complimentary, rather than competing approaches (we discuss this in for example in Turchin 2008 as well as Francois et al 2016). For the reasons detailed above, here our focus is on the macro-level patterns. We note this in the Materials and Methods section and have provided additional text in the Discussion section emphasizing that future work exploring more fine-grained data and qualitative case studies is needed to delve deeper into the broader patterns and processes suggested by the analyses here. 

Refs: 

Turchin, Peter. “Arise ‘Cliodynamics.’” Nature 454, no. 7200 (2008): 34–35.https://doi.org/10.1038/454034a.

François, Pieter, J. G. Manning, Harvey Whitehouse, Rob Brennan, T. E. Currie, Kevin Feeney, and Peter Turchin. “A Macroscope for Global History. Seshat Global History Databank: A Methodological Overview.” Digital Humanities Quarterly 10, no. 4 (2016). http://www.digitalhumanities.org/dhq/vol/10/4/000272/000272.html.

One final detail: the throwaway comment on naval warfare on line 69 isn’t adequate. 

This is a good point and we agree that this was a poorly articulated point. We have removed this ‘throwaway’ comment from the main text, which we believe reads more clearly now. We discuss our inclusion of primarily land-based, rather than specifically maritime, technologies in the SI (the Defining the Response Variable: coding military technologies in past societies section) which we feel is adequate to explain our approach to readers. We would however be willing to add a discussion on this in the main text, if that would be useful to readers. 

Responses to Reviewer 2 Comments

This paper examines the long-term evolution of military technologies using the Seshat: Global History Databank. Creating a series of composite variables, which use simpler variables as a proxy for more complex ones such as Information Complexity or Military Technologies, the authors then test several theoretical claims in the literature. Several factors seem to significantly predict the advancement of Military Technology (MilTech), e.g., global population size, cultural similarity, and the spread of iron and cavalry. They also find variables that do not appear to dramatically influence of the level of military technology. Specifically, with the notable exception of phylogeny, characteristics such as the scale and sophistication of a polity appear to be non-significant predictors. This leads them to conclude that military technology evolves, for the most part, as an exogenous variable.

Overall, I applaud the authors for doing a very thorough and ambitious investigation in the evolution of military technology. I believe the paper will provide a catalyst for many follow up studies and serve a vital role of stimulating scientific debate. I do, however, have two general points that I’d like to make.

We thank R2 for this positive and encouraging assessment. 

For my first point, I want to highlight a potential limitation of this approach, in that aggregation might mask a simpler explanation for the results of your model. That is, your predictors might be predicting only a subset of your 46 binary variables used to create your MilTech measure. One way to deal with this is to create alternative MilTech measures by randomly sampling from your set of 46 variables. What you do here is create multiple MilTech measures composed from different, randomly sampled subsets of your 46 variables. How much does this change your overall results in this study? It might be the case that what your model is predicting is a specific subset of technologies that disproportionately influence the overall result. If the result is not robust to these random subsets, then I believe it diminishes your claim that the variables are predicting military technology per se. Instead, it might be the case that your models are predicting a specific subset of the technologies, as opposed to an aggregate measure of overall military technology.

We addressed this point by the analysis and results described above (in our responses to the Editor). Note that randomly sampling from the 46 variables is not a feasible approach because of the way we construct our Warfare Characteristics (WCs). For example, if a society has iron technology, than we assume that it also has the knowhow to produce “simpler” technologies (copper, bronze). Similarly, if a society has crossbows, then it doesn’t matter whether they know atlatls. Thus, sampling randomly form the raw binary variables will not yield the desired test. Instead we implemented this suggestion by randomly sampling from the six WCs – please see the results above and the discussion of these results now included in the SI.

My second point relates to the variable for the existing stock of technologies. Here, you use the existing stock of technology as a proxy for the influence of current technological level on military technological evolution. This is done in two ways: as a temporal autoregressive process and focusing on horse riding and iron smelting. For the temporal autoregressive measure, you could also look at a related approach such as transfer entropy. The advantage of this is that it does not rely on using a single timeseries to predict its future state. Instead, you can use separate timeseries (X and Y), as predictors of one another. This tells you how much uncertainty is reduced in the future values of Y by knowing the past values of X given the past values of Y. It is a way of measuring the influence of one timeseries process (X) on another timeseries process (Y). I feel that this would help you disentangle the directionality of the relationship between some of your variables (as the measure is non-symmetric and X|Y does not equal Y|X). So, for instance, you could have the existing stock of technology as one timeseries and the miltech variable as another timeseries. You would predict that the information flow goes from the existing stock to miltech, but not necessarily the other way around.

This is a good point and we have improved our explanation of the logic underlying our dynamic regression methodology. The approach we use is known among the econometricians as ‘Granger Causality’, as was pointed out by the Editor above. Transfer Entropy (TE) is a closely related approach; in fact, it was shown to be the equivalent to Granger Causality for Gaussian variables (Barnett et al. 2009). There is some discussion in the literature about which approach is better to take. TE is a nonparametric method, and so it doesn’t depend on the assumption of linearity. However, we test for nonlinearities and include nonlinear effects, where detected, using polynomials. On the downside, nonparametric methods require more data and have lower statistical power when variables are well-behaved. In our case, both inspection of scatter plots and formal regression results suggest that in most cases linear forms capture well the relationships between the response and predictors. And where this is not the case (as with the functional relationship between MilTech and its lag), the quadratic relationship appears to capture the curvature well (which is not huge, as visual inspection suggests).

To summarize, we believe that our approach is doing precisely what the reviewer would like us to do – exploring the full range of possible cross-relationships between our data – and we feel that using TE would not add any additional information, but would reduce the statistical power of results. For these reasons, we have not included a TE analysis or discussion of this decision in the main text or as a supplementary analysis 

Ref: 

Barnett, L., A. B. Barrett and A. K. Seth (2009). "Granger Causality and Transfer Entropy Are Equivalent for Gaussian Variables." Physical Review Letters 103(23): 238701.

Finally, a minor point is that for the results in Table 1, you should probably use scientific notation for extremely small p-values (especially for the MilTech and MilTech.sq p-values where you just have 0). In fact, after writing this comment, I noticed you already did this in your supplementary materials (so it should be easy to address).

We thank R2 for pointing out this oversight. We have corrected this in Table 1 of the main text.

---

## [Editor Report · Decision Letter 1]

20 Sep 2021

Rise of the War Machines: Charting the Evolution of Military Technologies from the Neolithic to the Industrial Revolution

PONE-D-21-17737R1

Dear Dr. Hoyer,

We’re pleased to inform you that your manuscript has been judged scientifically suitable for publication and will be formally accepted for publication once it meets all outstanding technical requirements.

Kind regards,

Olivier Morin

Academic Editor

PLOS ONE
---

## [Editor Report · Acceptance letter]

27 Sep 2021

PONE-D-21-17737R1 

Rise of the War Machines: Charting the Evolution of Military Technologies from the Neolithic to the Industrial Revolution 

Dear Dr. Hoyer:

I'm pleased to inform you that your manuscript has been deemed suitable for publication in PLOS ONE. Congratulations! Your manuscript is now with our production department. 

Kind regards, 

on behalf of

Dr. Olivier Morin 

Academic Editor

PLOS ONE